# Classifying patients with depressive and anxiety disorders according to symptom network structures: A Gaussian graphical mixture model-based clustering

Jun Kashihara[1]*, Yoshitake Takebayashi[2], Yoshihiko Kunisato[3], Masaya Ito[4]

1 Department of Social Psychology, Faculty of Sociology, Toyo University, Tokyo, Japan, 2 Department of Health Risk Communication, School of Medicine, Fukushima Medical University, Fukushima, Japan, 3 Department of Psychology, School of Human Sciences, Senshu University, Kawasaki, Japan, 4 National Center for Cognitive Behavior Therapy and Research, National Center of Neurology and Psychiatry, Tokyo, Japan

* better.days.ahead1121@gmail.com

## Abstract

Patients with mental disorders often suffer from comorbidity. Transdiagnostic understandings of mental disorders are expected to provide more accurate and detailed descriptions of psychopathology and be helpful in developing efficient treatments. Although conventional clustering techniques, such as latent profile analysis, are useful for the taxonomy of psychopathology, they provide little implications for targeting specific symptoms in each cluster. To overcome these limitations, we introduced Gaussian graphical mixture model (GGMM)-based clustering, a method developed in mathematical statistics to integrate clustering and network statistical approaches. To illustrate the technical details and clinical utility of the analysis, we applied GGMM-based clustering to a Japanese sample of 1,521 patients ($M_{age}$ = 42.42 years), who had diagnostic labels of major depressive disorder (MDD; $n$ = 406), panic disorder (PD; $n$ = 198), social anxiety disorder (SAD; $n$ = 116), obsessive-compulsive disorder (OCD; $n$ = 66), comorbid MDD and any anxiety disorder ($n$ = 636), or comorbid anxiety disorders ($n$ = 99). As a result, we identified the following four transdiagnostic clusters characterized by i) strong OCD and PD symptoms, and moderate MDD and SAD symptoms; ii) moderate MDD, PD, and SAD symptoms, and weak OCD symptoms; iii) weak symptoms of all four disorders; and iv) strong symptoms of all four disorders. Simultaneously, a covariance symptom network within each cluster was visualized. The discussion highlighted that the GGMM-based clusters help us generate clinical hypotheses for transdiagnostic clusters by enabling further investigations of each symptom network, such as the calculation of centrality indexes.

## Introduction

Approximately forty-five percent of individuals with mental disorders suffer from comorbidity, or multiple mental disorders [1]. Comorbidity predicts poorer prognosis and health-related quality of life [2, 3]. There is a substantial clinical need to establish treatment guidelines

**Data Availability Statement:** The dataset is available via the Open Science Framework (OSF;

https://osf.io/6jnf4/), along with the preregistered data analysis plan and the R script.

**Funding:** JK, YT, YK, and MI were supported by Japan Society for the Promotion of Science KAKENHI (Grant numbers: 20K14171, 19K14419, 20K20870, and 17H04788, respectively,). MI was supported also by Japan Agency for Medical Research and Development Grant (Reference number: JP20dk0307084) and by National Center of Neurology and Psychiatry (NCNP) Intramural Research Grant (Reference number: 30-2). The funders had no role in study design, data collection and analysis, decision to publish, or preparation of the manuscript.

**Competing interests:** The authors have declared that no competing interests exist.

targeting comorbid cases (e.g., [4–6]). As frequently discussed in psychopathology studies (e.g., [7–9]), such high proportion of comorbidity result from traditional diagnostic criteria—such as the Diagnostic and Statistical Manual of Mental Disorders (DSM) [10] and the International Classification of Diseases (ICD) [11]—that do not fit the empirical data surrounding mental disorder symptoms. The lack of fit between traditional diagnostic criteria and empirical data results in low symptom specificity, marked diagnostic heterogeneity, and poor reliability, as well as high comorbidity [12–15]. Recent studies have begun to use data-driven approaches to understand psychopathology from a transdiagnostic (i.e., across-diagnostic; see [16] for detailed definitions of the term *transdiagnostic*) perspective (e.g., [7, 12, 17, 18]).

Studies that seek a transdiagnostic understanding of psychopathology often apply clustering techniques, which classify patients with mental disorders into several clusters based on quantitative symptom data. For example, latent class analysis (LCA) [19], which identifies several latent classes behind discrete symptom variables, has been used to derive subgroups within major depressive disorder [20] and postpartum depression [21]. Moreover, latent profile analysis (LPA) [22], which is an extension of LCA that uses continuous variables as indicators, has been used to identify latent classes among comorbid cases of psychopathology [23, 24]. These studies have frequently achieved a quantitative classification of mental disorders that is inconsistent with conventional diagnostic systems. For example, Kircanski et al. [24] identified latent classes characterized by i) mixed symptoms of anxiety and depression and by ii) intense symptoms of irritability, anxiety, depression, and attention deficit hyperactivity disorder (ADHD) using clinical data from children and adolescents.

Although conventional clustering techniques have provided a useful taxonomy of psychopathology, these techniques only identify each cluster's profile (e.g., Cluster 1 is characterized by intense symptoms of irritability, anxiety, depression, and ADHD). More importantly, these techniques do not provide direct treatment implications for each cluster (e.g., clinicians cannot judge treatment target priorities across symptoms of irritability, anxiety, depression, and ADHD). A technique that identifies not only symptom item mean scores but also rich information regarding symptom interplays in latent class estimations would help clinicians better understand cluster-specific symptom structures and identify key target symptoms in each cluster, both of which are indispensable for clinical reasoning and case formulation. Such a novel technique would become a gateway that connect clustering techniques and network approaches of psychopathology, which are rapidly developing to identify complex symptom interplays and calculate centrality indexes for each symptom [25–28].

To the best of our knowledge, no previous psychopathological (or psychological) studies have achieved the mixture of network estimation and cluster identification. Although Brusco et al. [29] have used a stepwise approach to analyze binary symptom data of depression and anxiety, in which they first performed *p*-median clustering [30, 31] and subsequently used so-called eLasso procedure [32] to explore the network structure within each cluster, they admit that such stepwise approaches can potentially bias the resulted network structures. The main aim of Brusco et al. [29] was to demonstrate the limitations of using network modeling without accounting for unobserved heterogeneity of participants; they noted that mixture modeling approaches should be developed in future to achieve more accurate estimations of networks within participant clusters.

In the field of computational statistics, on the other hand, a hybrid of LPA and cross-sectional network analysis has recently been developed [33]. As detailed later in this section, this hybrid analysis became possible after the recent development of model-based clustering—a popular framework in mathematical statistics for clustering multivariate data [34, 35]—and use of the Gaussian graphical mixture model (GGMM) [33, 36]. In this paper, we introduce the GGMM-based clustering technique with reference to Fop et al. [33], apply the technique to

an existing dataset of patients with depressive and anxiety disorders, and discuss the clinical utility of the techniques and future directions for integrating clustering and network approaches.

## GGMM-based clustering technique

Clustering techniques are statistical analyses that identify several groups within a multivariate dataset, and earlier techniques largely depended on heuristic procedures, including the Ward method [37] and $k$-means clustering [38]. As detailed in Fraley and Raftery [34] and McNicholas [35], model-based clustering is a counter movement of heuristic clustering. It assumes that multivariate data are generated by a finite mixture of a certain number of distributions and determines the number of clusters solely from quantitative criteria, such as the Bayesian Information Criterion (BIC) [39, 40]. The framework of model-based clustering includes many conventional clustering techniques: LCA [19], for example, can be understood as a latent class model-based clustering that assumes multivariate data arise from a mixture of discrete distributions. To cluster multivariate continuous data, the finite Gaussian mixture model (GMM) [34, 35, 41, 42], which assumes that data are generated by finite mixtures of multivariate normal (Gaussian) distributions, are widely used. As described in Fop et al. [33], the finite GMM estimates a mean vector and a covariance matrix for each identified cluster $k$ (for $1 \leq k \leq K$). The density of each data point in the finite GMM is given by:

$$f(\boldsymbol{x_i}|\boldsymbol{\Theta}) = \sum_{k=1}^{K} \tau_k \Phi(\boldsymbol{x_i}|\boldsymbol{\mu_k}, \boldsymbol{\Sigma_k}), \tag{1}$$

where $\boldsymbol{x_i}$ is the vector of observed variables, K is the fixed number of clusters, $\tau_k$ are mixing proportions for cluster $k$, $\Phi(\cdot)$ is the multivariate Gaussian density function, $\boldsymbol{\mu_k}$ is the mean vector for cluster $k$, $\Sigma_{\boldsymbol{k}}$ is the covariance matrix for cluster $k$, and $\boldsymbol{\Theta}$ is the vector of model parameters. The EM algorithm [43] is typically used to estimate the model with a fixed number of clusters K, and the most appropriate number for K is determined through model comparisons based on fit indexes, such as BIC [39, 40].

Finite GMM-based clustering can easily be over-parameterized when large numbers of variables are included in the model. Several remedies have been developed to deal with potential over-parametrization, such as eigenvalue decomposition [44, 45] and factorizing the covariance matrix [46]. However, these remedies do not fully consider the covariance matrix of original variables in cluster specification. Worse still, finite GMM-based clustering does not visualize interplays between variables within each identified cluster. To overcome such limitations, Fop et al. [33] developed GGMM-based clustering, which combines GMM-based clustering with the Gaussian graphical model (GGM) [47, 48]. In GGM, interplays between variables are visualized as a graph (i.e., network), $G = (V, E)$, where $V$ is the set of nodes (i.e., variables), and $E$ is the set of edges (i.e., associations between paired variables). The graph estimation is achieved through covariance matrices of variables with penalty terms, such as the graphical least absolute shrinkage and selection operator (GLASSO) procedure [49], in which all edge parameters are first estimated, then the small ones are shrunk to zero. In GGMM-based clustering, graph structures estimated in GGM are considered in identifying variable clusters. This is expressed in the following definition of each data point's density in the GGMM:

$$f(\boldsymbol{x_i}|\boldsymbol{\Theta}, \mathbb{G}) = \sum_{k=1}^{K} \tau_k \Phi(\boldsymbol{x_i}|\boldsymbol{\mu_k}, \boldsymbol{\Sigma_k}, G_k) \ \text{ with } \boldsymbol{\Sigma_k} \in C^+(G_k), \tag{2}$$

in which the graph for cluster $k$, $G_k$, the collection of graphs, $\mathbb{G}$, and the cone of positive definite matrices for the graphs, $C^+ (G_k)$ are simply added to formula (1) in the GMM.

GGMM-based clustering can be executed with the mixggm package [33] for R software. In this package, the structural EM algorithm [50, 51], which incorporates the structure learning step into the EM algorithm [43], is used to estimate the model with a fixed number of clusters K, and models with different numbers of clusters are compared using BIC [39, 40]. The penalty function used in the graph calculations is specified by the penalized likelihood approach [52, 53], and graphs with edges that represent covariances between nodes are calculated as a result.

### The present study

To demonstrate the clinical and statistical implications of GGMM-based clustering, we re-analyzed an existing dataset of 1,521 Japanese patients with depressive and anxiety disorders (see the Methods section for details). Depressive and anxiety disorders are the most comorbid clinical conditions among mental disorders [54–56]. Therefore, we expected that this dataset, which included comorbid participants with depressive and anxiety disorders, was suitable for demonstrating the discrepancies (or similarities) between latent classes obtained by GGMM-based clustering and conventional diagnostic labels, and for simultaneously visualizing complex symptom interplays in comorbid disorders.

## Materials and methods

### Dataset

We used an existing dataset collected for a large research project launched in Japan that aimed to validate several measures assessing depressive and anxiety symptoms and related constructs, such as emotion regulation skills [57–60]. The minimal dataset needed to replicate our results is freely available via the Open Science Framework (OSF; https://osf.io/6jnf4/). Collection of the data used in this study was approved by the National Center of Neurology and Psychiatry Institutional Review Board (Approval number: A2013-022; Title of the project: An online survey using clinical and non-clinical samples to validate the Overall Anxiety/Depression and Impairment Scales [OASIS/ODSIS]). In this project, an online survey was conducted in January 2014, using an internet marketing research company's panelist pool in Japan (Macromill Inc; https://group.macromill.com/). When the survey was conducted, 1,095,443 panelists were registered in this pool, and 389,265 of them were labeled as "disease panelists" based on their self-reported clinical status in February 2013. A total of 2,459 Japanese anonymous disease panelists 18 years or older participated in the survey, and their labels were as follows: major depressive disorder (MDD; $n = 619$), panic disorder (PD; $n = 619$), social anxiety disorder (SAD; $n = 576$), obsessive-compulsive disorder (OCD; $n = 645$). In addition, data from "non-disease panelists" ($n = 371$) were collected as counterparts of those from disease panelists.

Since these diagnostic labels were based on self-descriptions made about one year before the study, a series of items designed to check panelists' current diagnostic status were used during the survey (e.g., "Are you currently diagnosed as having major depressive disorder and being treated for the problem in a medical setting?"). According to the responses to these items, a total of 2,830 participants, including both disease and non-disease panelists, were divided into the following categories: MDD ($n = 406$), PD ($n = 198$), SAD ($n = 116$), OCD ($n = 66$), comorbid MDD and any anxiety disorder ($n = 636$), comorbid anxiety disorders ($n = 99$), other mental disorders ($n = 146$), and non-clinical ($n = 1,163$). The analyses used only the data from the 1,521 participants (775 female, 746 male; $M_{age} = 42.42$, $SD = 9.50$) with one or more depressive or anxiety disorder diagnoses. In other words, we excluded the data from participants categorized as other mental disorders ($n = 146$) or non-clinical ($n = 1,163$)

from the analyses. As the online survey required the participants to respond to all the items, the resulting dataset included no missing values.

## Measures

We selectively used the MDD, PD, SAD, and OCD symptom data for the present study. For detailed information on measures included in the dataset, see [59, 60].

**MDD symptoms.** MDD symptoms were assessed using the Japanese version [61] of the Patient Health Questionnaire [62]. This 9-item measure asks participants about the frequency of depressive symptoms over the past two weeks, using a 4-point Likert scale from 1 (Not at all) to 4 (Nearly every day). Doi et al. [57] found that this Japanese measure has the following two-factor structure: i) cognitive/affective symptoms (6 items; e.g., "feeling down, depressed, or hopeless") and ii) somatic symptoms (3 items; e.g., "poor appetite or overeating").

**PD symptoms.** PD symptoms were assessed using the Japanese version [60] of the Anxiety Sensitivity Index-3 [63]. This 18-item measure assesses participants' anxiety sensitivity, which is defined as fear of arousal-related physical and psychological sensations [64, 65], using a 5-point Likert scale from 1 (very little) to 5 (very much). Ebesutani et al. [66] found that these 18 items reflect a general factor of anxiety sensitivity plus the following three subfactors: i) physical concerns (6 items; e.g., "It scares me when my heart beats rapidly"), ii) cognitive concerns (6 items; e.g., "When I cannot keep my mind on a task, I worry that I might be going crazy"), and iii) social concerns (6 items; e.g., "When I tremble I fear what people might think of me").

**SAD symptoms.** SAD symptoms were assessed using the Japanese version [67] of the Fear of Negative Evaluation Scale-Short Form [68]. This 12-item measure assesses participants' tendency to feel threatened by the prospect of negative evaluation from others, using a 5-point Likert scale from 1 (not at all) to 5 (very likely). Sasagawa et al. [67] showed that these 12 items exhibit a high internal consistency and have a one-factor structure. Sample items included "I worry about what other people will think of me even when I know it doesn't make any difference."

**OCD symptoms.** OCD symptoms were assessed using the Japanese version [69] of the Obsessive-Compulsive Inventory-Short Form [70]. This 18-item measure assesses participants' obsessive and compulsive symptoms using a 5-point Likert scale from 1 (not at all) to 5 (very much). Koike et al. [71] showed that the Japanese version has the following six-factor structure: i) hoarding (3 items; e.g., "I have saved up so many things that they get in the way"), ii) checking (3 items; e.g., "I check things more often than necessary"), iii) ordering (3 items; e.g., "I get upset if objects are not arranged properly"), iv) neutralizing (3 items; e.g., "I feel I have to repeat certain numbers"), v) washing (3 items; e.g., "I sometimes have to wash or clean myself simply because I feel contaminated"), and vi) obsessing (3 items; e.g., "I find it difficult to control my own thoughts").

## Data analysis

The data analysis plan was preregistered on the OSF (https://osf.io/6jnf4/), and R script used in the analyses are freely available there. First, we conducted a series of confirmatory factor analyses to calculate factor scores that reflected the factor structure for each symptom measure (see the Measures subsection for details). Second, using the mixggm package [33] for R software, we conducted GGMM-based clustering with the default setting (i.e., tuning parameter β was set to 0) to i) determine the number of clusters that can be obtained from MDD, PD, SAD, and OCD symptom data; ii) summarize a profile of symptom factor scores for each cluster; iii) evaluate the concordance between the cluster allocation of patients and their diagnostic labels;

and iv) visualize interplays of individual symptoms as a covariance network for each cluster. Third, using the qgraph [72] and bootnet [27] packages for R, we estimated a partial correlation network of symptoms for each cluster and computed centrality indexes, including bridge ones, for each symptom. To estimate the partial correlation networks, we used the GLASSO procedure [49] with Extended BIC (EBIC) [73]. Finally, using the networktools package [74] for R software, we computed bridge centrality indexes in each cluster to specify symptoms that connect different disorders.

As explained on OSF (https://osf.io/6jnf4/), the data analysis procedures noted above were slightly changed from those planned *a priori*. First, when conducting GGMM-based clustering, we entered factor scores instead of individual symptom items. This change was made after we found that GGMM-based clustering could not achieve a convergence when using items with ordinal scales and that continuous variables need to be entered. Second, we estimated partial correlation networks and calculated centrality indexes in addition to estimating covariance networks computed in GGMM-based clustering. We expected that partial correlation networks and centrality indexes (i.e., strength, closeness, and betweenness) would be more informative and potentially reveal richer clinical implications than covariance networks alone. Although closeness (i.e., the inverse of the sum of geodesic distances from one node to the other nodes) and betweenness (i.e., number of times one node lies in the shortest paths between other nodes) centrality were widely used in previous investigations of psychopathology networks (e.g., [75–77]), Bringmann et al. [78] recently argued that these two indexes are unstable in psychopathology networks and do not fit common assumptions underlying psychological research. We, therefore, reported closeness and betweenness centrality as supplementary information and interpret only strength centrality (i.e., the sum of absolute values of edge weights connected to each node).

We also have some notes regarding the availability of the R package we used. Although we have conducted GGMM-based clustering on 12 February 2021 using the mixggm package, that package has been removed from the CRAN repository (https://cran.rstudio.com/web/packages/mixggm/index.html) on 21 April 2021 by the developers. To increase the transparency of our analyses, we have uploaded the users' manual of the mixggm package, which has been initially provided by the developers via the CRAN repository, on OSF (https://osf.io/6jnf4/).

## Results

### GGMM-based clustering

The results of a series of confirmatory factor analyses on symptom measures were detailed in the R Markdown file on OSF (https://osf.io/6jnf4/), and the factor loadings obtained were used to calculate factor scores of MDD, PD, SAD, and OCD symptoms. Correlations between factor scores included in GGMM-based clustering are displayed in Fig 1. The mixGGM function built in the mixggm package [33] revealed that GGMM-based clustering with a five-cluster solution could not be properly calculated since the variance and covariance matrices became not positive definite within one or more clusters. Furthermore, it automatically compared BIC [39, 40] across GGMMs with one- to four-cluster solutions and then returned the results on the four-cluster solution with the lowest BIC. The participants' cluster membership is presented in Table 1, and symptom score factor profile for each cluster is displayed in the left panel of Fig 2. As shown, Cluster 4 ($n$ = 791; 52.0% of the participants), the largest cluster, was characterized by strong symptoms of all four disorders. By contrast, Cluster 3 ($n$ = 138; 9.1% of the participants) was characterized by weak symptoms of all four disorders. Cluster 1 ($n$ = 235; 15.5% of participants) was characterized by strong OCD and PD symptoms and moderate

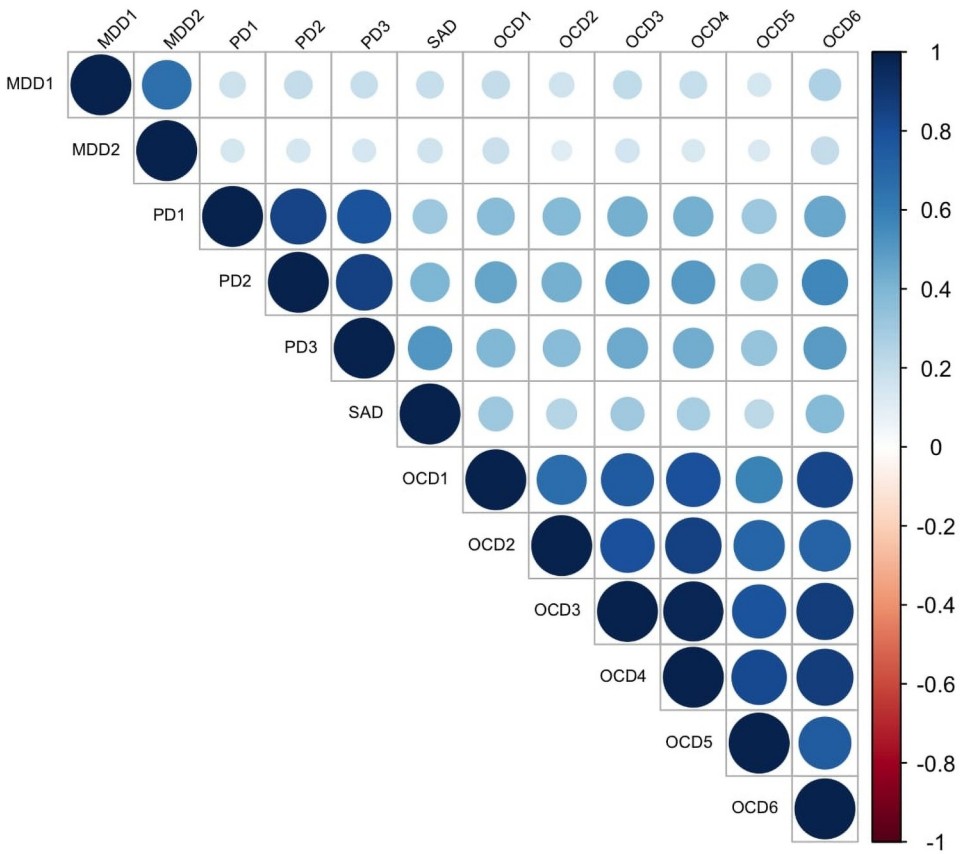

**Fig 1. Correlations between factor scores of depressive and anxiety symptoms.** MDD = major depressive disorder, PD = panic disorder, SAD = social anxiety disorder, OCD = obsessive-compulsive disorder. MDD1 = cognitive/ affective symptoms, MDD2 = somatic symptoms, PD1 = physical concerns, PD2 = cognitive concerns, PD3 = social concerns, OCD1 = hoarding, OCD2 = checking, OCD3 = ordering, OCD4 = neutralizing, OCD5 = washing, OCD6 = obsessing.

MDD and SAD symptoms, and Cluster 2 (*n* = 357; 23.5% of participants) was characterized by moderate MDD, PD, and SAD symptoms and weak OCD symptoms. Of note, GGMM cluster allocation was inconsistent with participants' diagnostic labels. As shown in Table 1, all clusters included a certain number of participants with each diagnostic label.

The covariance symptom networks, which were estimated simultaneously with the cluster allocation, are displayed in the right panel of Fig 2. The four symptom network shapes differed

**Table 1. Participants' allocations to the Gaussian graphical mixture model (GGMM)-based clusters (*N* = 1,521).**

|  | Cluster 1 | Cluster 2 | Cluster 3 | Cluster 4 | Total |
|---|---|---|---|---|---|
| Major depressive disorder (MDD) | 61 | 125 | 51 | 169 | 406 |
| Panic disorder (PD) | 20 | 63 | 35 | 80 | 198 |
| Social anxiety disorder (SAD) | 15 | 45 | 9 | 47 | 116 |
| Obsessive-compulsive disorder (OCD) | 2 | 6 | 6 | 52 | 66 |
| Comorbid MDD and any anxiety disorder | 120 | 101 | 33 | 382 | 636 |
| Comorbid anxiety disorders | 17 | 17 | 4 | 61 | 99 |
| Total | 235 | 357 | 138 | 791 | 1,521 |

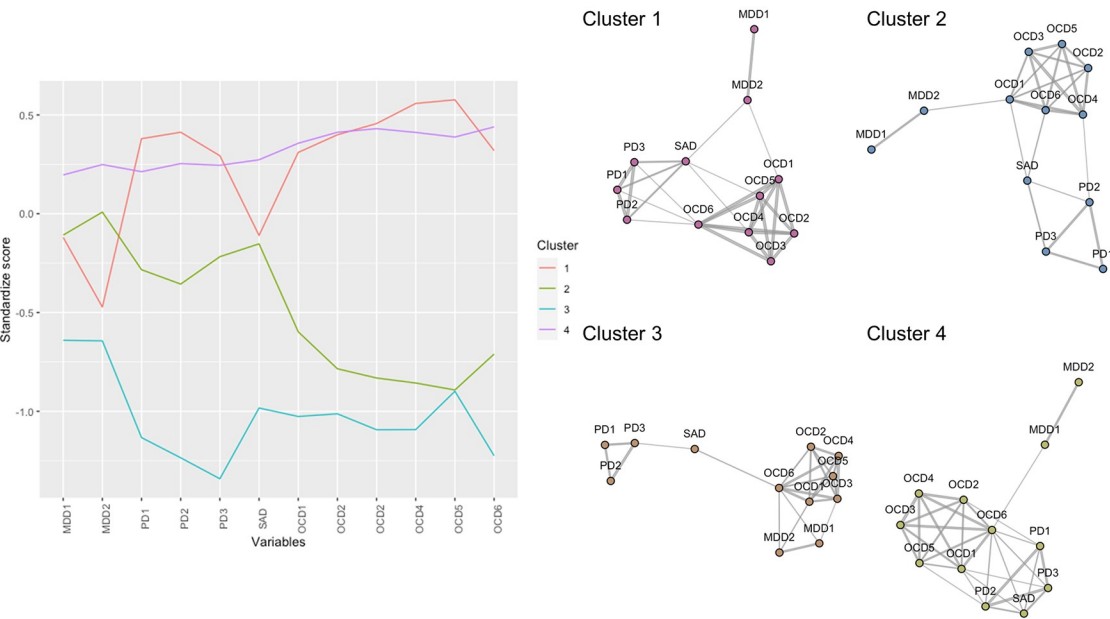

**Fig 2. The symptom-profile of patient clusters identified by GGMM (left panel) and the covariance symptom network estimated for each cluster (right panel).** The thickness of edges in the symptom networks reflects the values of covariances. MDD = major depressive disorder, PD = panic disorder, SAD = social anxiety disorder, OCD = obsessive-compulsive disorder. MDD1 = cognitive/affective symptoms, MDD2 = somatic symptoms, PD1 = physical concerns, PD2 = cognitive concerns, PD3 = social concerns, OCD1 = hoarding, OCD2 = checking, OCD3 = ordering, OCD4 = neutralizing, OCD5 = washing, OCD6 = obsessing.

markedly from each other, and several features could be identified from visual inspection across the networks. For example, OCD symptoms were strongly interconnected in each cluster. SAD symptoms were characteristically located near the center of each network, except the Cluster 4 network, which had strong symptoms for all four disorders. Moreover, MDD symptoms were relatively isolated (i.e., located far apart from anxiety-related symptoms and directly connected with a small number of symptoms) in every symptom network except Cluster 3, which had low levels of symptomatology.

## Partial correlation network estimations via GLASSO regulation with EBIC

The partial correlation network for each cluster identified in GGMM-based clustering is displayed in Fig 3. Several marked features of Cluster 2 and 4 networks can be identified from visual inspections. These networks consisted of both positive and negative edges (i.e., partial correlations), and there were negative edges even within the community of OCD symptoms. By contrast, the Cluster 1 and 3 networks consisted only of positive edges. It is also noteworthy that larger numbers of nonzero edges were obtained for Clusters 2 and 4, compared to Clusters 1 and 3. To summarize, network structures were much more complex in Clusters 2 and 4 than in Clusters 1 and 3.

Centrality indexes for each symptom in each cluster are summarized in Fig 4. Interestingly, the OCD neutralizing symptom (OCD4) had the highest strength in each cluster. It is also noteworthy that MAD and SAD symptoms had relatively low strength in each cluster.

## Bridge centrality indexes calculations

Bridge centrality (i.e., a variant of centrality that takes communities of nodes into account), including expected influence (i.e., the sum of raw values of edge weights connected to each

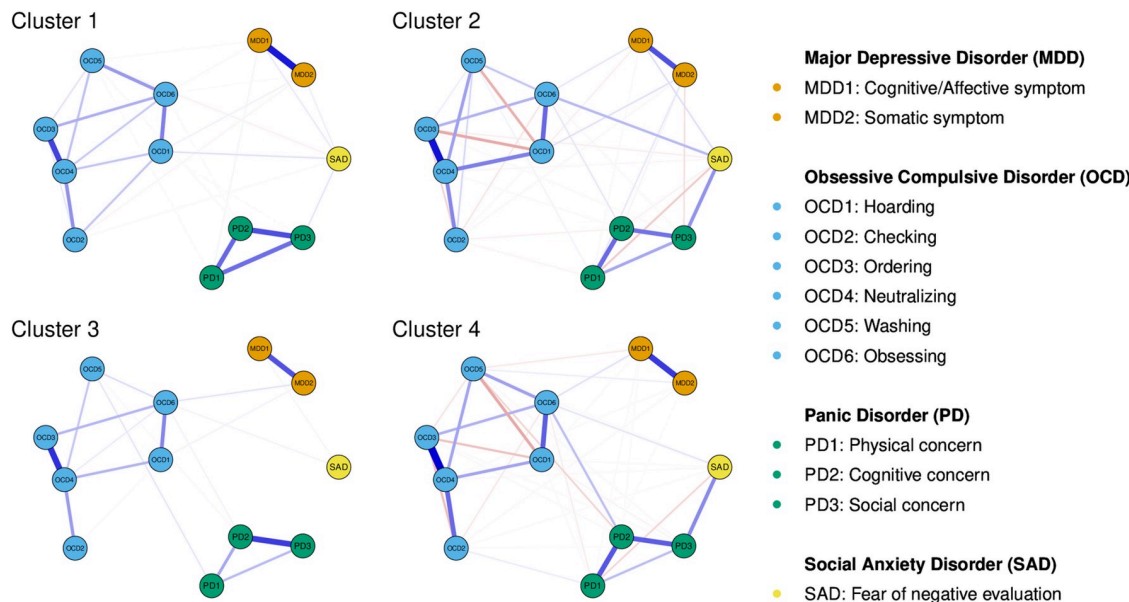

**Fig 3. Partial correlation network estimated for each patient cluster.** Nodes represent symptoms, and edges represent partial correlations between them. The thickness of an edge reflects the absolute value of the regularized partial correlation. Blue and red edges represent positive and negative regularized partial correlations, respectively.

node), for each symptom in each cluster is summarized in Fig 5. As shown, SAD symptoms had the nearly highest bridge strength and expected influence in every network. Of interest, the physical concerns of PD (PD1), as well as SAD symptoms, had the nearly highest bridge strength and expected influence in Cluster 4, with strong symptoms of all four disorders.

## Discussion

### Clinical utility of GGMM-based clustering

In this study, we introduced GGMM-based clustering—developed by Fop et al. [33] in the mathematical statistics field—in the context of psychopathology research. Using an existing dataset of patients with depressive and anxiety disorders, our application demonstrated several lines of clinical utility of GGMM-based clustering. First, GGMM-based clustering can identify several transdiagnostic clusters (e.g., Cluster 1 with strong OCD and PD symptoms and moderate MDD and SAD symptoms; see the left panel of Fig 2). Second, GGMM-based clustering can visualize the symptom network in each cluster simultaneously with cluster identification. We believe that such a mixture of clustering and network approaches has the potential to advance transdiagnostic understanding of psychopathology. In fact, from the right panel of Fig 2, we can identify several network structure similarities and differences across the identified clusters (e.g., symptoms of OCD strongly interconnected in each cluster; MDD symptoms were relatively isolated in each cluster, except Cluster 3 with low levels of symptomatology). By capturing such network structure information, one can move beyond conventional clustering techniques that show only each cluster's profile to detailed network descriptions for each cluster.

Third, and most importantly, GGMM-based clustering serves as a primer to further investigate cluster characteristics from a network perspective. Since GGMM-based clustering takes graph structures of variables into account in identifying clusters, one can safely conduct

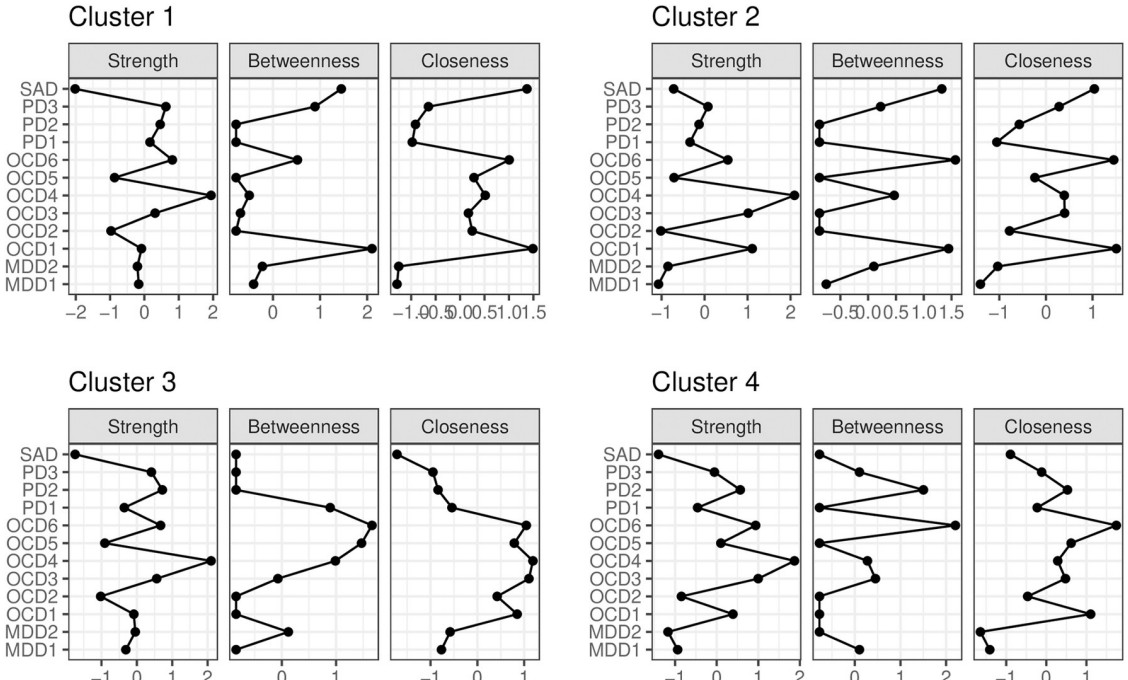

**Fig 4. Centrality indexes (standardized *z*-scores) calculated in each partial correlation network.** MDD = major depressive disorder, PD = panic disorder, SAD = social anxiety disorder, OCD = obsessive-compulsive disorder. MDD1 = cognitive/affective symptoms, MDD2 = somatic symptoms, PD1 = physical concerns, PD2 = cognitive concerns, PD3 = social concerns, OCD1 = hoarding, OCD2 = checking, OCD3 = ordering, OCD4 = neutralizing, OCD5 = washing, OCD6 = obsessing.

intensive network-based analyses that can potentially guide clinical hypotheses, including estimation of partial correlation networks and calculation of centrality indexes. In fact, we can elicit several clinical implications from such intensive analyses conducted in this study: we can i) assume that we need to carefully identify target symptoms when intervening patients categorized in Cluster 2 and 4 cases, which include both positive and negative edges (see Fig 3); ii) judge the neutralizing symptom of OCD (OCD4) with the highest strength as the target symptom in every identified cluster (see Fig 4); and iii) hypothesize that SAD symptoms, with nearly the highest bridge strength and expected influence, sustain the comorbid networks in every cluster (Fig 5). These clinical implications are useful for developing new transdiagnostic treatment for patients with each cluster. We believe that such intensive investigations cannot be adequately achieved with a simple combination of conventional clustering techniques and intensive network analyses, because it is theoretically inconsistent to impose network structures on the clusters that are identified without accounting for the interplays among items.

## Statistical implications of GGMM-based clustering

The clinical utility of GGMM-based clustering was achieved by a statistical integration of clustering and network approaches. Such integration is relatively new in the network sciences, and most of the previous integration has been achieved to identify clusters of nodes within an estimated network [79, 80]. By contrast, the techniques to describe a network within each of the identified clusters are scarce in previous studies of psychopathology networks (for a review, see [81]), and the necessity of introducing mixture modeling with accounting for unobserved heterogeneity has been discussed [29]. The GGMM-based clustering introduced here fills such

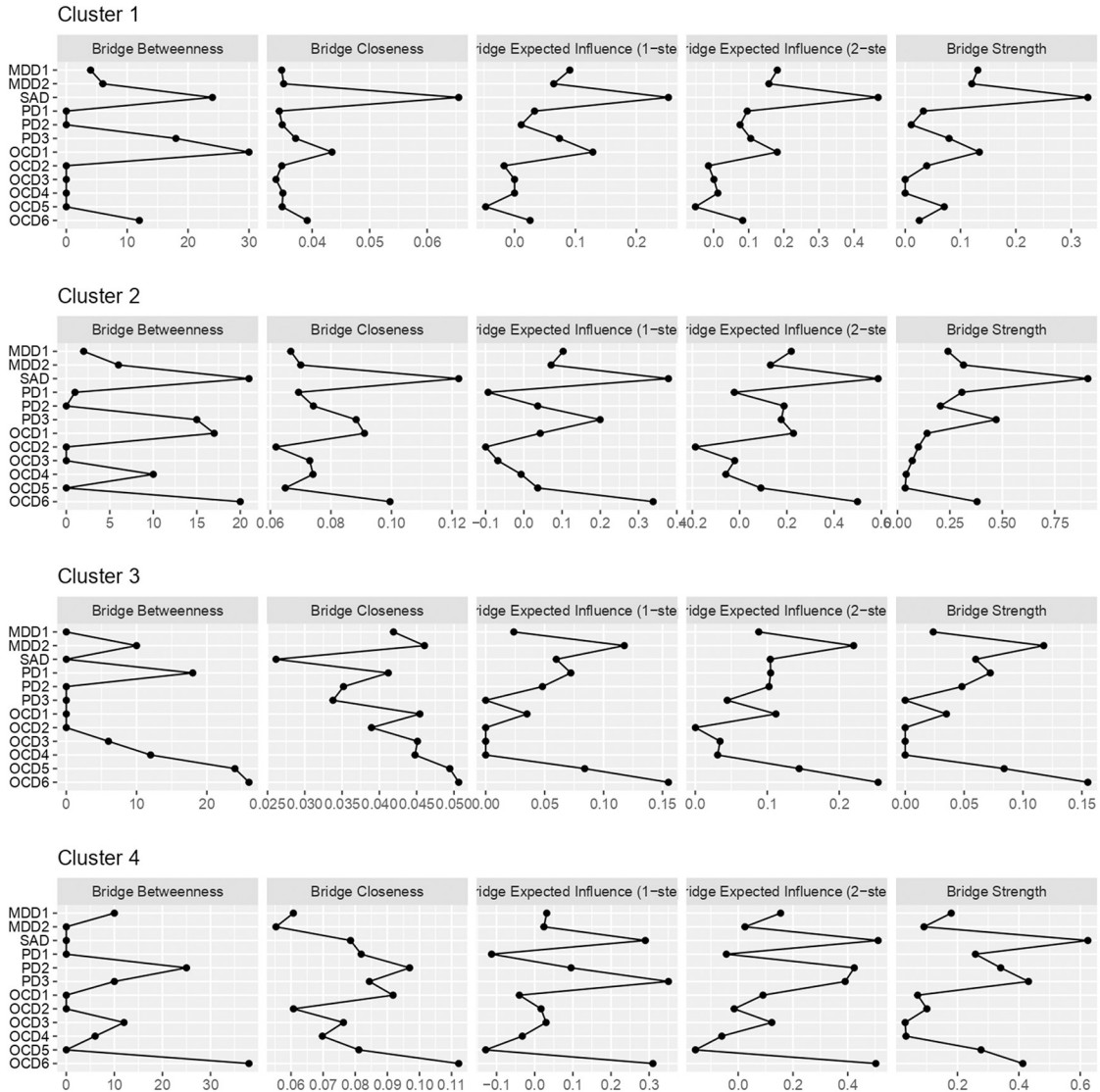

**Fig 5. Bridge centrality indexes (standardized *z*-scores) calculated in each partial correlation network.** MDD = major depressive disorder, PD = panic disorder, SAD = social anxiety disorder, OCD = obsessive-compulsive disorder. MDD1 = cognitive/affective symptoms, MDD2 = somatic symptoms, PD1 = physical concerns, PD2 = cognitive concerns, PD3 = social concerns, OCD1 = hoarding, OCD2 = checking, OCD3 = ordering, OCD4 = neutralizing, OCD5 = washing, OCD6 = obsessing.

a blank area in network sciences and may lead to further development of statistical models and tools to investigate networks within clusters.

More broadly, GGMM-based clustering can be regarded as a successful integration of latent variable and network models, because the cluster is an exemplar of latent variables. Although the network approach is frequently discussed in contrast to the latent variables approach (e.g., [75, 82–84]), these approaches are mathematically equivalent (i.e., a latent variable model can be converted to a network model with the same free parameters [85–88]). These models, therefore, are open to integration. Some integrated models and analyses—including the residual network model [89] and the exploratory graph analysis used for estimating the dimensionality of psychological constructs [90]—have recently been developed. We hope that the GGMM-

based clustering will accelerate such innovative integrations of latent variable and network models.

## Limitations and future directions

GGMM-based clustering and the mixggm package [33] have several limitations that should be noted when developing future research directions. First, the current mixggm package visualizes only the structure of a network within each cluster and provides neither partial correlations between nodes nor centrality indexes, both of which have been widely investigated in cross-sectional analyses of psychopathology networks. To increase the clinical utility of GGMM-based clustering and promote clinical hypothesizing, future studies should either develop new packages of GGMM-based clustering or revise the analysis itself to enable partial correlation network and centrality index estimation. Developing such statistical tools would help users determine whether each edge is positive or negative and identify target and bridge symptoms. Second, in our study, GGMM-based clustering could be conducted not with individual symptom items but with symptom factor scores. We hope that future studies will identify why and how often estimation errors in GGMM-based clustering occur with single items and provide remedies for such estimation errors. Such studies will lead to further development of techniques to investigate networks within clusters and increase options for users.

There were also limitations regarding the application study design. First, because our data were collected through an online panel survey, our results might be affected by biases endemic in internet-based clinical research, such as inflation in reported symptoms [91], and participants' carelessness or deliberate fraud in item response [92]. Second, the diagnostic labels included in the dataset were based on participants' self-report on single items (e.g., "Are you currently diagnosed as having major depressive disorder and being treated for the problem in a medical setting?") and, therefore, may be less accurate than those based on structured clinical interviews or well-validated screening tools. For a more accurate transdiagnostic understanding of depressive and anxiety disorders, future studies should use rigorous inclusion criteria to replicate the present application study. Third, our dataset included limited numbers of diagnostic labels (i.e., MDD, PD, SAD, OCD); therefore, our findings cannot be generalized to the entire network of depressive and anxiety disorders. To depict a full picture of comorbidity in depressive and anxiety disorders, future research should include all diagnostic labels for these two disorders, or curate empirical studies of network approaches that include some of the diagnostic labels from these two disorders. The latter approach may become possible in the near future, considering the increasing number of empirical studies investigating networks of: i) MDD and generalized anxiety disorder [93–98], ii) MDD and OCD [99], and iii) MDD and posttraumatic stress disorder [100].

Despite these limitations, our application study successfully demonstrated both the clinical utility and statistical implications of GGMM-based clustering. We hope that GGMM-based clustering will promote a better understanding of comorbid mental disorder structures and that further integrations of clustering and network approaches will be achieved in the future.

## Acknowledgments

We would like to thank Editage by Cactus Communications Inc. (www.editage.com) for English language editing. Portions of the present study were presented as a poster at the 84th Annual Convention of the Japanese Psychological Association held online (September–November 2020).

## Author Contributions

**Conceptualization:** Jun Kashihara, Yoshitake Takebayashi, Yoshihiko Kunisato.

**Data curation:** Yoshitake Takebayashi, Yoshihiko Kunisato, Masaya Ito.

**Formal analysis:** Yoshitake Takebayashi, Yoshihiko Kunisato.

**Funding acquisition:** Jun Kashihara, Yoshitake Takebayashi, Yoshihiko Kunisato, Masaya Ito.

**Methodology:** Jun Kashihara, Yoshitake Takebayashi, Yoshihiko Kunisato.

**Project administration:** Jun Kashihara.

**Resources:** Masaya Ito.

**Software:** Yoshitake Takebayashi, Yoshihiko Kunisato.

**Supervision:** Yoshitake Takebayashi, Yoshihiko Kunisato, Masaya Ito.

**Visualization:** Yoshitake Takebayashi, Yoshihiko Kunisato.

**Writing – original draft:** Jun Kashihara.

**Writing – review & editing:** Jun Kashihara, Yoshitake Takebayashi, Yoshihiko Kunisato, Masaya Ito.

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
