## [Decision Letter · Decision Letter 0]

15 Jul 2021

PONE-D-21-17750

Classifying patients with depressive and anxiety disorders according to symptom network structures: A Gaussian graphical mixture model-based clustering

PLOS ONE

Dear Dr. Kashihara,

Thank you for submitting your manuscript to PLOS ONE. After careful consideration, we feel that it has merit but does not fully meet PLOS ONE’s publication criteria as it currently stands. Therefore, we invite you to submit a revised version of the manuscript that addresses the points raised during the review process.

We look forward to receiving your revised manuscript.

Kind regards,

Zezhi Li, Ph.D., M.D.

Academic Editor

PLOS ONE

Reviewers' comments:

Reviewer's Responses to Questions

**Comments to the Author**

1. Is the manuscript technically sound, and do the data support the conclusions?

Reviewer #1: Yes

Reviewer #2: Yes

2. Has the statistical analysis been performed appropriately and rigorously? 

Reviewer #1: Yes

Reviewer #2: N/A

3. Have the authors made all data underlying the findings in their manuscript fully available?

Reviewer #1: Yes

Reviewer #2: No

4. Is the manuscript presented in an intelligible fashion and written in standard English?

Reviewer #1: Yes

Reviewer #2: Yes

5. Review Comments to the Author

Reviewer #1: The authors classified patients with depressive and anxiety disorders using a Gaussian graphical mixture model-based clustering method. They identified four transdiagnostic clusters. This study provided a better understanding of comorbid mental disorder structures. The authors also mention some limitations to be improved in the future.

Some minor comments:

1. In the abstract, panic disorder (PD; n = 193), however, in the table 1 “n = 198”. The authors need to explain why the two numbers are inconsistent.

2. In line 49 and 56. The authors need to describe the example instead of “e.g., [4–6]”, “(e.g., [7–9])”, and “(e.g., [7,12,18,19])”.

3. In the introduction section, the authors describe too much about GGMM-based clustering technique, which need to be placed to method section.

4. In figure 1, the legend ranged from -1 to 0 is unnecessary, should be remove.

5. The name of the x-axis should be added to Figure 4 and 5 to make them easier for readers to understand.

6. The authors should minimize citations about clustering methods and algorithms (too redundant and unnecessary!!!), and adopt more MDD-related recently published papers. For example, 1） Major Depressive Disorder: Advances in Neuroscience Research and Translational Applications. Li Z, Ruan M, Chen J, Fang Y. Neurosci Bull. 2021 Jun;37(6):863-880. 2) The association of clinical correlates, metabolic parameters, and thyroid hormones with suicide attempts in first-episode and drug-naïve patients with major depressive disorder comorbid with anxiety: a large-scale cross-sectional study. Zhou Y, Ren W, Sun Q, Yu KM, Lang X, Li Z, Zhang XY. Transl Psychiatry. 2021 Feb 4;11(1):97.

7. Language needs to be polished and refined.

Reviewer #2: This study applies the Gaussian graphical mixture model (GGMM)-based clustering to detect mental disorders comorbidities.

Please see the comments below.

1-Dataset section; please describe your dataset social-demographic variables (age, sex group,...etc.) and missing value, how to deal with them.

2- Please indicate in a table; Psychiatric disorders and Co-morbidities with count (n) and percentage (% ) instead of explaining all in contexts.

3- please make clear; GGMM-based clustering parameters tuning has been used.

4- Please, if possible, list the ICD 9 - 10 codes of all diagnoses in the supplementary section; to better perceive related comorbidities.

6. PLOS authors have the option to publish the peer review history of their article (what does this mean?). If published, this will include your full peer review and any attached files.

Reviewer #1: No

Reviewer #2: No

---

## [Author Response · Author response to Decision Letter 0]

6 Aug 2021

■Response to Reviewer #1

Thank you for your constructive comments, which have helped us to improve our manuscript. We truly appreciate the time and energy you have expended.

In the revised version of the manuscript, we have addressed your comments and suggestions as follows:

Comment #1:

In the abstract, panic disorder (PD; n = 193), however, in the table 1 “n = 198”. The authors need to explain why the two numbers are inconsistent.

Reply #1:

This was a typing error in the Abstract, and “n = 198” was the correct number for participants with panic disorder. Thank you very much for your careful reading.

Comment #2:

In line 49 and 56. The authors need to describe the example instead of “e.g., [4–6]”, “(e.g., [7–9])”, and “(e.g., [7,12,18,19])”.

Reply #2:

Although we understand your point, we put more emphasis on logical flow between paragraphs than on details within paragraphs. As the first paragraph is meant to introduce the context but not the content of data-driven psychopathology, it seems not advisable to describe each example in too much detail. We would appreciate it if you would understand our style.

Comment #3:

In the introduction section, the authors describe too much about GGMM-based clustering technique, which need to be placed to method section.

Reply #3:

Although we respect your specialty and understand your tastes, our orientation differs from yours. Our main targets are psychometricians and clinical psychologists who are interested in advanced statistical models, and our main objective was to demonstrate the significance of GGMM-based clustering to such readers. Therefore, we need to explain how GGMM-based clustering is differentiated from other conventional techniques in the Introduction section.

On the basis of the above, we decided not to revise this point. We hope that you understand our orientation.

Comment #4:

In figure 1, the legend ranged from -1 to 0 is unnecessary, should be remove.

Reply #4:

Although we understand your point, this style of legend is set as a default in the R package and is unchangeable by users. Therefore, we have no choice but to use the legend, ranging from −1 to +1. Moreover, we do not think that the lower half of the legend is unnecessary. It reminds readers that correlation coefficients can generally take negative values, and the absence of red circles in the graph area will impress readers that there were no negative correlations between the factor scores.

Comment #5:

The name of the x-axis should be added to Figure 4 and 5 to make them easier for readers to understand.

Reply #5:

As in the legend of Figure 1 (see Reply #4), the styles of Figures 4 and 5 are set as a default in the R package and are unchangeable by users. Therefore, we revised the figure titles to indicate that each x-axis represents standardized z-scores of the centrality indexes (see lines 317 and 332). Thank you for your constructive comments.

Comment #6:

The authors should minimize citations about clustering methods and algorithms (too redundant and unnecessary!!!), and adopt more MDD-related recently published papers. For example, …

Reply #6:

As noted in Reply #3, our main targets are psychometricians and clinical psychologists who are interested in advanced statistical models. As these citations and algorithms are indispensable to satisfy such readers, we decided not to minimize them. It should also be noted that the editorial policy of PLOS ONE emphasizes rigorous methodology and accurate reporting of the procedures. To satisfy this editorial policy, it is advisable to maintain the current form.

In addition, the phrase “too redundant and unnecessary!!!” seems quite inappropriate as a review comment. Please be logical and constructive, instead of emotionally criticizing our orientation. As highlighted in standard publication manuals (e.g., APA and AMA manuals), respect for others’ orientations is the foundation of academic writing and sound peer review.

Moreover, we are wondering why you have recommended additional citations of recently published papers regarding MDD. We have already cited a number of such papers. Could you explain why you have picked those two papers, which are seemingly from the same group, out of the global literature on MDD? If those two papers are from your group, and you cannot logically explain why they need to be cited, we cannot follow your suggestion to save your dignity. Such self-serving behavior by reviewers is regarded as a violation of research ethics in the current scientific community and is sometimes condemned via anonymous platforms, such as PubPeer.

Comment #7:

Language needs to be polished and refined.

Reply #7:

As noted in the Acknowledgments section, we used a professional English editing service (Editage by Cactus Communications Inc.) before submitting. Therefore, we wondered what points you thought would need to be improved. We also think that detailed grammatical matters should be handled by journal editorial staff rather than peer reviewers. Could you specify the points, if any, that need to be polished and that cannot be handled by journal editorial staff?

We wish to thank you again for your constructive comments. We look forward to hearing from you soon.

Sincerely,

Jun Kashihara, Ph.D.

Department of Social Psychology, Faculty of Sociology, Toyo University. 

5-28-20 Hakusan, Bunkyo-ku, Tokyo 112-8606, Japan

E-mail: better.days.ahead1121@gmail.com

 

■Reply to Reviewer #2

Thank you for your constructive comments, which have helped us to improve our manuscript. We truly appreciate the time and energy you have expended.

In the revised version of the manuscript, we have addressed your comments and suggestions as follows:

Comment #1:

Dataset section; please describe your dataset social-demographic variables (age, sex group,...etc.) and missing value, how to deal with them.

Reply #1:

Although you might not have noticed it, social demographics have already been reported in the second paragraph of the Dataset subsection (see line 177). As the data from 1,521 out of 2,830 participants were analyzed, we first explained why and how 1,521 participants were selected and then described their demographics. Considering that previous papers have reported the demographics of all 2,830 participants, we did not repeat them in the current manuscript to avoid confusing readers.

Regarding the missing values, we added the following sentence at the end of the Dataset subsection (see lines 179–181): “As the online survey required the participants to respond to all the items, the resulting dataset included no missing values.”

Comment #2:

Please indicate in a table; Psychiatric disorders and Co-morbidities with count (n) and percentage (% ) instead of explaining all in contexts.

Reply #2:

Although this seems to be an option, we decided not to provide percentages in Table 1 to maintain visibility. Considering that Table 1 has six rows and is already wide in its current form, it would become “too much” if we added percentages. As visibility directly leads to interpretability, it is not advisable to decrease visibility.

Comment #3:

please make clear; GGMM-based clustering parameters tuning has been used.

Reply #3:

On the basis of your comment, we have declared that GGMM-based clustering was conducted with the default setting, in which the tuning parameter was set to 0, in the Data Analysis subsection (see line 226).

Comment #4:

Please, if possible, list the ICD 9 - 10 codes of all diagnoses in the supplementary section; to better perceive related comorbidities.

Reply #4:

Although this seems to be an option, we decided not to provide supplementary material. As readers today can easily access the ICD and DSM criteria, it does not seem necessary to replicate them here. Our policy is to keep papers concise, and we would appreciate it if you would understand it.

We wish to thank you again for your constructive comments. We look forward to hearing from you soon.

Sincerely,

Jun Kashihara, Ph.D.

Department of Social Psychology, Faculty of Sociology, Toyo University.

5-28-20 Hakusan, Bunkyo-ku, Tokyo 112-8606, Japan

E-mail: better.days.ahead1121@gmail.com

---

## [Decision Letter · Decision Letter 1]

18 Aug 2021

Classifying patients with depressive and anxiety disorders according to symptom network structures: A Gaussian graphical mixture model-based clustering

PONE-D-21-17750R1

Dear Dr. Kashihara,

We’re pleased to inform you that your manuscript has been judged scientifically suitable for publication and will be formally accepted for publication once it meets all outstanding technical requirements.

Kind regards,

Zezhi Li, Ph.D., M.D.

Academic Editor

PLOS ONE

Additional Editor Comments (optional):

Reviewers' comments:

Reviewer's Responses to Questions

**Comments to the Author**

1. If the authors have adequately addressed your comments raised in a previous round of review and you feel that this manuscript is now acceptable for publication, you may indicate that here to bypass the “Comments to the Author” section, enter your conflict of interest statement in the “Confidential to Editor” section, and submit your "Accept" recommendation.

Reviewer #1: All comments have been addressed

Reviewer #2: All comments have been addressed

2. Is the manuscript technically sound, and do the data support the conclusions?

Reviewer #1: Yes

Reviewer #2: Yes

3. Has the statistical analysis been performed appropriately and rigorously? 

Reviewer #1: Yes

Reviewer #2: N/A

4. Have the authors made all data underlying the findings in their manuscript fully available?

Reviewer #1: Yes

Reviewer #2: Yes

5. Is the manuscript presented in an intelligible fashion and written in standard English?

Reviewer #1: Yes

Reviewer #2: Yes

6. Review Comments to the Author

Reviewer #1: (No Response)

Reviewer #2: Dear Editor,

Thank you for considering me as a reviewer for this publication in your journal. I checked the authors' responses and completely satisfied with them. I recommend this paper for publication.

Thank you kindly.

7. PLOS authors have the option to publish the peer review history of their article (what does this mean?). If published, this will include your full peer review and any attached files.

Reviewer #1: No

Reviewer #2: No

---

## [Editor Report · Acceptance letter]

24 Aug 2021

PONE-D-21-17750R1 

Classifying patients with depressive and anxiety disorders according to symptom network structures: A Gaussian graphical mixture model-based clustering 

Dear Dr. Kashihara:

I'm pleased to inform you that your manuscript has been deemed suitable for publication in PLOS ONE. Congratulations! Your manuscript is now with our production department. 

Kind regards, 

on behalf of

Dr. Zezhi Li 

Academic Editor

PLOS ONE